# WISH YOU WERE HERE: HINDSIGHT GOAL SELECTION FOR LONG-HORIZON DEXTEROUS MANIPULATION

**Todor Davchev**[1,2,†*], **Oleg Sushkov**[1,†], **Jean-Baptiste Regli**[1], **Stefan Schaal**[3], **Yusuf Aytar**[1], **Markus Wulfmeier**[1], **Jon Scholz**[1]

[1]DeepMind, [2]University of Edinburgh, [3]Intrinsic LLC

## ABSTRACT

Complex sequential tasks in continuous-control settings often require agents to successfully traverse a set of "narrow passages" in their state space. Solving such tasks with a sparse reward in a sample-efficient manner poses a challenge to modern reinforcement learning (RL) due to the associated long-horizon nature of the problem and the lack of sufficient positive signal during learning. Various tools have been applied to address this challenge. When available, large sets of demonstrations can guide agent exploration. Hindsight relabelling on the other hand does not require additional sources of information. However, existing strategies explore based on task-agnostic goal distributions, which can render the solution of long-horizon tasks impractical. In this work, we extend hindsight relabelling mechanisms to guide exploration along task-specific distributions implied by a small set of successful demonstrations. We evaluate the approach on four complex, single and dual arm, robotics manipulation tasks against strong suitable baselines. The method requires far fewer demonstrations to solve all tasks and achieves a significantly higher overall performance as task complexity increases. Finally, we investigate the robustness of the proposed solution with respect to the quality of input representations and the number of demonstrations.

## 1 INTRODUCTION

Recent advances in model-free reinforcement learning (RL) have enabled successful applications in a variety of practical, real-world tasks (Gu et al., 2017; Levine et al., 2018; Riedmiller et al., 2018; OpenAI et al., 2019; Sharma et al., 2020; Gupta et al., 2021). Given the complexity of these tasks, additional information such as demonstrations often plays an important role (Vecerik et al., 2019; Zuo et al., 2020; Davchev et al., 2020; Sutanto et al., 2020; Luo et al., 2021). These methods are particularly useful to robotics as they enable efficient learning with only sparse rewards, which are easier to define. However, scaling such solutions to more complex long-horizon sequential settings with limited number of demonstrations remains an open challenge (Gupta et al., 2020).

Solving long-horizon dexterous manipulation tasks is an important problem as it enables a wide range of useful robotic applications ranging from object-handling tasks, common in collaborative settings, through to contact-rich dual arm manipulations, often seen in manufacturing. Consider the bi-manual 3.5mm jack cable insertion problem as illustrated in Figure 1. Solving such tasks with vanilla *demo-driven RL* often fails due to the difficulty of assigning credit over long horizons in the presence of noisy exploration and sparse rewards.

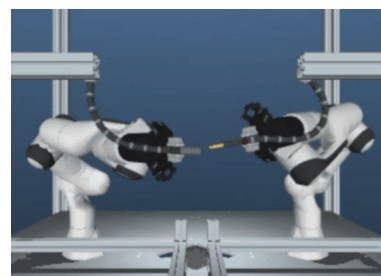

Figure 1: HinDRL solving a 3.5mm jack cable insertion task.

Self-supervision mechanisms like Hindsight Experience Replay (HER) (Andrychowicz et al., 2017) offer an alternative to ease the sparse reward problem by providing a stronger learning signal through additional goal-reaching tasks which are generated from the agent's trajectories. While the more frequent reward can be beneficial, the agent's trajectory

---

*Work done during an internship at DeepMind; † Denotes equal contribution.

distribution is often considerably more complex and non-stationary in comparison to the target-task distribution. When the final task is very complex and the main task reward is not perceived, this can lead to learning a capable, goal-reaching policy for states close to the agent's initial states while unable to complete the actual task.

Our contribution is based on the insight that demonstration states can be viewed as samples of the target task distribution, and can therefore be used to constrain the self-supervision process to task-relevant goals. We introduce a framework for *task-constrained* goal-conditioned RL that flexibly combines demonstrations with hindsight relabelling. Unlike HER, which learns a general goal-conditioned agent, we train a goal-conditioned agent specialized at achieving goals which directly lead to the task solution. The approach further allows to smoothly vary the task relevance of the relabelling process. Unlike conventional goal-conditioned RL, we enable agents to solve tasks through utilising abstract goal formulations, such as inserting a 3.5mm jack, common in the context of complex sequential tasks. We achieve this through using a continuously improving target goal distribution for the online goal selection stage. We refer to our method as a Hindsight Goal Selection for Demo-Driven RL or HinDRL for short. We demonstrate that the proposed solution can solve tasks where both demonstration-driven RL and its self-supervised version with HER struggle. Specifically, we show that HinDRL can dramatically reduce the number of required demonstration by an order of magnitude on most considered tasks.

## 2 RELATED LITERATURE

Using demonstrations in RL is a popular tool for robot learning. Classic approaches, such as (Atkeson & Schaal, 1997; Peters & Schaal, 2008; Tamosiunaite et al., 2011), use expert demonstrations to extract good initial policies before fine-tuning with RL. Kim et al. (2013); Chemali & Lazaric (2015) use demonstrations to learn an imitation loss function to bootstrap learning. However, those solutions are not applied to complex sequential tasks. Follow up work that use deep neural networks enable the application of demo-driven RL to more complex tasks such as Atari(Aytar et al., 2018; Pohlen et al., 2018; Hester et al., 2018), or robotics (Vecerik et al., 2017; 2019; Paine et al., 2018). We build upon DPGfD(Vecerik et al., 2019), which we explain in more details in Section 3.

The Atari literature often requires large number of environment steps (Salimans & Chen, 2018; Bacon et al., 2017). This can result in hardware wear and tear if applied on a physical robotics system. Alternatively, a number of robotics solutions assume the existence of a manually defined structure, e.g. through specifying an explicit set of primitive skills (Stulp et al., 2012; Xie et al., 2020), or a manually defined curricula (Davchev et al., 2020; Luo et al., 2021). However, hand-crafting structure can be sub-optimal in practice as it varies across tasks. In contrast, restricting the search space by greedily solving for specific goals (Foster & Dayan, 2002; Schaul et al., 2015) can be combined with implicitly defined curricula through retroactive goal selection (Kaelbling, 1993; Andrychowicz et al., 2017). Hindsight goal assignment has been successfully applied to multi-task RL (Li et al., 2020; Eysenbach et al., 2020), reset-free RL (Sharma et al., 2021) but also for batch RL (Kalashnikov et al., 2021; Chebotar et al., 2021) and hierarchical RL (Wulfmeier et al., 2020; 2021). However, these approaches explore based on task-agnostic goal distributions and compensate with additional policy structure such as allowing for longer training to concurrently learn a hierarchy, or using large amounts of offline data. In this work, we extend hindsight relabelling mechanisms to guide exploration along task-specific distributions implied from a limited number of demonstrations and study its performance against strong task-agnostic goal distributions.

Combining hindsight relabelling with demonstrations is not a novel concept. Ding et al. (2019) relabels demonstrations in hindsight to improve generative adversarial imitation learning. The proposed solution is used to learn a goal-conditioned reward function that can be applied on demo-driven RL for hindsight relabelling. Gupta et al. (2020) propose a hierarchical data relabelling mechanism that uses demonstrations to extract goal-conditioned hierarchical imitation-based policies and apply them to multi task RL. Nair et al. (2018a) fuses demo-driven RL similar to us but uses HER's final-goal sampling mechanism and combines it with structured resets to specific demo states to overcome the difficulties of exploration for long-horizon tasks, which would be difficult to achieve in a continuous robotics task in practice. Zuo et al. (2020) uses demonstrations to bootstrap TD3 and combines it with HER. However, in all those works, hindsight goals were always chosen directly from the agent's own trajectory. Instead, we *consistently select goals directly from a task distribution implied from a collection of successful demonstrations*. Hindsight goal selection has previously been generalized via adopting representation learning. Florensa et al. (2019) applied HER

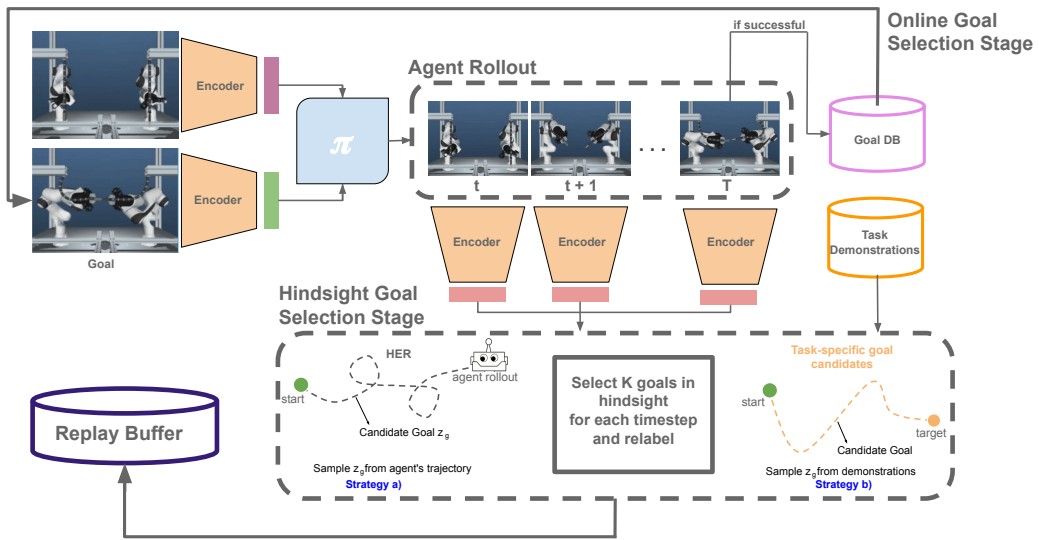

Figure 2: The HinDRL pipeline performing a 3.5mm jack cable insertion. The input to the policy are encoded target goal and the current robot state. At train time, the produced episodes are relabelled in hindsight using a choice of sampling strategies. We consider two main strategies for sampling goals in hindsight. Strategy a) shows the standard hindsight goal selection strategies that select goals from states within the trajectory being relabelled; and strategy b) shows a mechanism that focuses on sampling goals directly from some collection of successful trajectories. The resulted relabelled data is fed into the replay buffer.

to a pixels-to-torque reaching task where sample efficiency was not a direct constraint. Nair et al. (2018b) used a $\beta-$balanced variational auto encoder ($\beta$-VAE) to enable learning from visual inputs with a special reward. While $\beta$-VAE is broadly a powerful tool for unsupervised representation learning, it does not take into account the temporal nature of sequential robotics tasks. This makes them sub-optimal in the context of our work (Chen et al., 2021). In contrast, contrastive-learning based techniques such as temporal cycle consistency (TCC) (Dwibedi et al., 2019) can provide more informative representations as previously discussed in concurrent work (Zakka et al., 2022). In this work we combine HinDRL, a framework for goal-conditioned demo-driven RL with both learnt and engineered representations and provide a comprehensive sensitivity analysis to the quality of the encoder and the dependency against the number of demonstrations used.

## 3 PRELIMINARIES

Consider a finite-horizon discounted Markov decision process (MDP), $\mathcal{M} = (S, A, P, r, \gamma, T)$ with transition probability $P : S \times A \times S \mapsto [0, 1]$. Let the current state and goal $s, g \in S \subseteq \mathbb{R}^{n_s}$ be elements in the state space $S$, and $a \in A \subseteq \mathbb{R}^{n_a}$ denote the desired robot action. We define a sparse environmental reward $r(s)$ that assigns 0/1 reward only at the final state of an episode. Let $\psi(\cdot)$ be an encoding function that embeds a given state and goal to a latent state $z = \psi(s)$ and a latent goal $g = \psi(g)$. Let $\zeta = (\mathbf{x}_{t_0}, \dots, \mathbf{x}_T)$ be a trajectory with a discrete horizon $T$, a discount function $\gamma(\cdot)$, a state-action tuple $x_t = (s_t, a_t, g)$ and a trajectory return $R(\zeta) = \sum_{t=t_0}^{T} \gamma r(s_t)$. In this setting, a transition $(s_t, a_t, s_{t+1}, g, r = 0)$ can be 'relabelled' as $(s_t, a_t, s_{t+1}, \hat{g} \approx s_{t+1}, r = 1)$ and both the original and relabelled transitions can be used for training. A separate, goal conditioned reward $r(z_{t+1}, \hat{z}_g) = \mathbb{1}[z_{t+1} = \hat{z}_g]$ uses the latent state and goal and assigns reward during relabelling. Finally, we also assume access to D demonstration trajectories $\mathcal{D} = \{\zeta_j\}_{j=0}^{D}$ that reach the goals $\{\hat{g}_j\}_{j=1}^{D}$ retrieved from the final state of the demonstration. Then, a goal-conditioned deterministic control policy parameterised by $\theta$, $\pi_\theta(a|z, z_g)$ selects an action $a$ given a latent state $z$ and goal $z_g$. In this context, we define an optimal policy $\pi^* = \arg\max_{\pi \in \bar{\pi}} J(\pi)$, where $J(\pi) = \mathbb{E}_{s_0 \sim p(s_0), g \sim \pi(g)}[R(\zeta)]$.

DPGfD(Vecerik et al., 2019) is a powerful algorithm that bootstraps learning through a BC loss applied directly to the actor. This loss is typically defined as the L2 distance between the predicted by the policy actions and the true actions of a demonstration and is decayed proportionally to the number of environmental steps. Gradual conversion from using a BC loss to using the actor-critic loss is typically done using Q filtering or gradually decaying the BC loss. We employ the latter. The algorithm uses an L2 regularised actor-critic loss with a distributional critic, introduced in (Bellemare

et al., 2017). Using BC loss with DDPG was recently applied to a goal-conditioned setting (Ding et al., 2019) but goal-conditioned policy learning was never used with demo-driven distributional RL.

# 4 METHODOLOGY

This section describes our method, HinDRL. First, we extend DPGfD to its goal-conditioned interpretation and introduce a mechanism that deals with intermediate goals. Next, we introduce two demonstration-driven goal sampling strategies and discuss how to use them for goal-conditioned policy learning. Finally, we discuss how to extract temporally consistent representations and define a distance-based goal-conditioned reward function. We summarise our framework in Figure 2.

## 4.1 GOAL-CONDITIONED DEMO-DRIVEN REINFORCEMENT LEARNING

Similar to DPGfD we combine reinforcement and imitation learning. We utilise a data set of demonstrations, $\mathscr{D} = \{(s_0, a_0, s_g, \ldots)_t^k\}_{k=1}^D$ which are used to seed a replay buffer for RL, and for supervised training of the policy. We used both 6D Cartesian velocity and a binary open/close actions, which we modelled using a standard regression for the predicted velocity and a classification for the binary prediction. We use encoded state $z$ and encoded goal $z_g$ as input to the policy network,

$$\mathscr{L}_{BC} = ||\pi(z, z_g)_c - a_c||_2^2 - \sum_{i=1}^{2} a_b^o log(\pi(z, z_g)_b^o). \tag{1}$$

The BC loss is applied only to transitions from successful trajectories.

For RL we utilized the standard deterministic policy-gradient (DPG, Silver et al. (2014)) loss for the deterministic actions (velocities), and stochastic value-gradient (SVG, Heess et al. (2015)) for the stochastic actions (gripper positions). We used the Gumbel-Softmax trick (Jang et al., 2016; Maddison et al., 2016) to reparameterize the binary actions for the SVG loss. Following Vecerik et al. (2019) we applied both the BC and RL losses to train the policy, and annealed the BC loss throughout training to allow the agent to outperform the expert.

We use a distributional critic (Bellemare et al., 2017), modeling the Q-function with a categorical distribution between 0 and 1 (with 60 evenly spaced bins). The learning loss is the KL-divergence between our current estimate and a projection of the 1-step return on the current bins. Therefore, the loss becomes,

$$\mathscr{L}_{TD} = KL(Q(z_t, a_t, z_g)||\Phi(r_t + \gamma(t) * Q_{target}(z_t, \pi(z_t, z_g), z_g))), \tag{2}$$

where $\Phi$ is an operator that projects a distribution on the set of bins.

## 4.2 GOAL SELECTION FROM DEMONSTRATIONS

Typically, goal-conditioned policy learning has two separate stages for goal selection (Andrychowicz et al., 2017): an online stage where a policy is conditioned on a specific goal during a policy rollout, and a hindsight goal selection stage where the produced trajectory is being relabelled in hindsight (See Figure 2). While Andrychowicz et al. (2017) only consider the original goal in the online stage, we describe below how this perspective can be extended to increase robustness.

**Online goal selection** A target goal of a sequential task, such as inserting a plug into a socket, is in principle more abstract than the physical skill of reaching a specific configuration. That is, being an $\varepsilon$ distance away from a target configuration might still result in a failed insertion. We mitigate this issue by maintaining a separate goal database, $G_{db}$, comprised of final states over successful executions. We bootstrap this database with the final states from $\mathscr{D}$ but we also continuously grow $G_{db}$ as learning progresses. Conceptually, the more examples of final goal states we collect, the better understanding of the target goal of the current task we will have. In this context, for every episode we sample a new $\hat{z}_g$ from a distribution over the set of all possible goals in $G_{db}$. We denote this as $R_G = \{z_g \in G_{db} : p(z_g) > 0\}$ and we refer to this process as online goal-conditioning. In this work, we always store the resulted $\hat{z}_g$-conditioned trajectory, $\zeta$, in the replay buffer. As in classic RL, the last step of $\zeta$ is rewarded using the environment reward $r(s_t)$.

**Hindsight goal selection** In contrast, hindsight goal selection is the process of retroactively sampling candidate goal states from some goal distribution $p(z_g)$ after an episode rollout. Here, $z_g$ are not conceptually related to the abstract formulation of a target task as $\hat{z}_g$ is. Instead, they represent different stages from the behaviour that results in solving the abstract task. The way this stage works is we sample new goals for each time step of the trajectory $\zeta$ using some candidate goal distribution

---

**Algorithm 1** HinDRL

1: Initial $\theta$, $\phi$, $\mathscr{D} = \{\tau^k\}_{k=1}^D$, $RB = \emptyset$
2: Policy $\pi_\theta$, encoder $\psi_\phi$, $p(s_0)$
3: // if trainable, learn encoder offline here
4: // encode demos, add to replay buffer
5: $RB \leftarrow RB \bigcup \mathscr{D}$
6: // relabel trajectories and add to RB
7: $RB \leftarrow RB \bigcup \{\text{generate\_samples}(\tau^k, \mathscr{D})\}_{k=1}^D$
8: // add final state from demos to goal db
9: $G_{db} = G_{db} \bigcup \{\tau_T^k\}_{k=1}^D$
10: **for** iter $i \in iters_{max}$ **do**
11: $\quad$ $s_0 \leftarrow p(s_0)$ // initial state
12: $\quad$ $\hat{g} \sim G_{db}$ // target goal
13: $\quad$ // trajectory $\zeta = \{(z_t, a_t, z_{t+1}, r_t, z_g)\}_{t=1}^T$
14: $\quad$ $\zeta \leftarrow \text{rollout}(s_0, g, \pi(\cdot), \psi(\cdot))$
15: $\quad$ **if** *success* **then**
16: $\quad\quad$ // if successful, store last state in goal db
17: $\quad\quad$ $G_{db} \leftarrow G_{db} \bigcup z_T$ // $z_T \in \zeta$
18: $\quad$ **end if**
19: $\quad$ $RB \leftarrow RB \bigcup \zeta$ // add rollout to RB
20: $\quad$ // relabel $\zeta$ and add to RB
21: $\quad$ $RB \leftarrow RB \bigcup \text{generate\_samples}(\zeta, \mathscr{D})$
22: $\quad$ // optionally, retrain $\phi$ here
23: $\quad$ $\theta \leftarrow \text{train\_}\pi(R)$
24: **end for**

---

**Algorithm 2** Generate samples

1: Input: trajectory, $\zeta$, task demos $\mathscr{D}$
2: $\chi \leftarrow \emptyset$ // relabelled transitions
3: **for** sampler $\in$ sampling strategies list **do**
4: $\quad$ data $\leftarrow \emptyset$ // support data
5: $\quad$ **if** sampler.name is Rollout-conditioned **then**
6: $\quad\quad$ data $= \zeta$ // HER
7: $\quad$ **else if** sampler.name is Task-conditioned **then**
8: $\quad\quad$ data $= \mathscr{D}$ // Task
9: $\quad$ **end if**
10: $\quad$ // compose support for goal distribution
11: $\quad$ $R_G = \{z_g \in data : p(z_g) > 0\}$
12: $\quad$ // sample new goals, $T = |\zeta|$
13: $\quad$ $\{z_g \sim p(z_g)\}_{t=1}^T$
14: $\quad$ // store relabelled transitions using Eq. 3
15: $\quad$ $\chi \leftarrow \chi \bigcup \{(z_t, a_t, z_{t+1}, r(z_{t+1}, z_g), z_g)\}_{t=1}^T$
16: **end for**
17: Returns: $\chi$

---

Figure 3: Algorithm for Hindsight goal selection for Demo-driven Reinforcement learning (HinDRL)

$p(z_g)$ implied from some given behaviour. Typically we sample candidate goals from the future states in rollout trajectories. Then, we re-evaluate the given transition with the newly sampled goal and assign a new reward. This is achieved with the help of a goal conditioned reward $r(z_t, z_g)$ and not the environment reward from the previous paragraph. Thanks to the goal-conditioned formulation, fitting a Q function from both of these sources of reward is well-posed. In this section we propose a way for composing a hindsight goal distribution $p(z_g)$, that is targeted on the specific task at hand.

Candidate goals can be acquired through self-supervision (as in HER), where a goal distribution is comprised of states from the agent's trajectory $\zeta$. The agent retrospectively samples feasible goals to 'explain' its own behaviour under the current goal-conditioned policy $\pi$. However, similar to concurrent work (Gupta et al., 2020; Pertsch et al., 2020) we observed that choosing goals from $p(z_g|\zeta)$ does not work well for long-horizon tasks.

However, hindsight goal selection does not need to be constrained to the same trajectory. Goals can be sampled directly from a task distribution, implied from a set of successful trajectories, such as demonstrations. Therefore, selecting task-specific goals can come from $p(z_g)$ with support $R_G = \{z_g \in \mathscr{D} : p(z_g) > 0\}$. We used a uniform distribution over the demonstration states. However, similar to HER, this can benefit from alternative formulations too. In practice, the support of the task-distribution, $R_G$, can be enriched over time with additional positive task completions from the training process too. However, we do not do it in this work.

Using the agent's trajectory $\zeta$ is useful when modelling the agent's behaviour. However, it can struggle to scale beyond normal-horizon tasks, Gupta et al. (2020). On the other hand, using demo-driven samplers can speed up training by directly modelling the task distribution. However, it requires certain level of confidence over the quality of the task representation. These two support distributions can complement each other and do not need to be disjoint. Joining both types of trajectories can be particularly useful in cases where using just demonstration states can fail to make the sparse reward problem easier, e.g. when the representations fail to capture the notion of progress. We provide an ablation against joint-condition samplers in Appendix A.3, including using the union, $\zeta \bigcup \mathscr{D}$, and intersection $\zeta \bigcap \mathscr{D}$ of the two sets of goals. Our results indicate that doing so can improve the performance of HinDRL with representations that do not preserve the notion of progress and speed up training for complex tasks (see Appendix A.4 and A.5).

### 4.3 TIME-CONSISTENT REPRESENTATIONS

Synthesising goal-conditioned policies from raw state observations can be impractical in high dimensional spaces. Using raw states can not only deteriorate the speed of learning (Hessel et al.,

2018), but it also makes it difficult to define a comprehensive goal-conditioned reward for relabelling. Instead, we encode our state observations into a lower dimensional latent space $\mathbb{R}^{n_z}$, using an encoder $\psi$ to obtain a latent state $z_t = \psi(s_t)$ and a latent goal $z_g = \psi(g)$. Next, we propose both expert-engineered and learnt time-consistent representations.

**Engineered representations** In the presence of an expert, low dimensional representations can be programmed to include features that bear a notion of progress, e.g. through measuring the distance from a target goal, as well as the notion of contact, such as whether an insertion was successful. These notions vary across each task and we provide additional details for each of the engineered encoders in Appendix A.9.

**Learnt representations** We propose as an alternative to the engineered encoding, a self-supervised learnt representation that can be directly used with the provided demonstrations and refined over time. We ensure consistent and aligned notion of episode progress in our learnt representation without the need of labels by employing a differentiable cycle-consistency loss (TCC)(Dwibedi et al., 2019). This allows us to learn a representation by finding correspondences across time between trajectories of differing length, effectively preserving the notion of progress.

Training of TCC involves encoding an observation $o_t$ from trajectory $U$ to a latent $z_t^U$, and checking for cycle-consistency with another trajectory $V$ that is not necessarily of the same length as $U$. Cycle consistency is defined via nearest-neighbor – if the nearest embedding in $V$ to $z_t^U$ is $z_m^V$, i.e. $nearest(z_t^U, V) = z_m^V$, then $z_t^U$ and $z_m^V$ are cycle-consistent if and only if $nearest(z_m^V, U) = z_t^U$. Learning this requires a differentiable version of cycle-consistency measure, such as regression-based or classification-based one. In this work, we employ the latter.

We additionally evaluate a non-temporal embedding using a $\beta$-VAE, similar to (Nair et al., 2018b) (see Appendix A.5).

## 4.4 DISTANCE-BASED REWARD SPECIFICATION

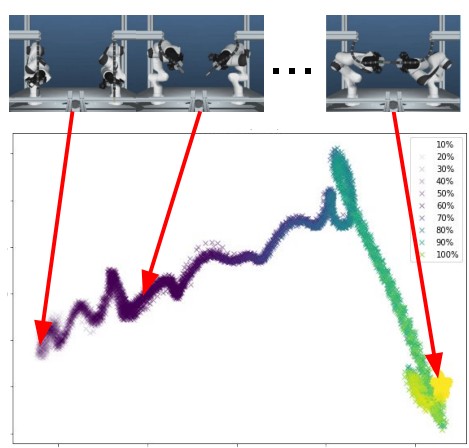

Figure 4: t-SNE of latent representations obtained with TCC (Dwibedi et al., 2019) against novel successful trajectories. Temporal color-coding, purple is start and yellow is goal.

Considering local smoothness in the learnt goal representation space, we use a distance-based goal-conditioned reward function. This allows us to increase the number of positive rewards and therefore ease the hard exploration problem. We particularly use,

$$r(z, z_g) = \mathbb{1}[||z - z_g|| < \varepsilon], \qquad (3)$$

for some threshold $\varepsilon$, where latent state $z$ and latent goal $z_g$ are considered similar if the $l_2$ distance between them is below $\varepsilon$. We provide more details on how we obtain the threshold in Appendix A.8. Figure 4 illustrates the t-SNE visualisation of a temporally aligned representation space learnt with TCC. The notion of progress in TCC is consistent across all 100 plotted trajectories. We provide additional details in Appendix A.5. TCC is inherently capable of temporally aligning different in length and motion trajectories that pass through similar stages. This makes it particularly suitable to use with distance-based metrics like Eq. 3.

The final proposed algorithm, together with the relabelling technique, is detailed in Algorithm 1.

## 5 EXPERIMENTS

We are interested in answering the following questions: i) is task-conditioned hindsight goal selection useful for long-horizon sequential tasks; ii) does it improve over DPGfD and HER's sample efficiency; iii) is HinDRL a demonstration efficient solution and how does it perform in a few-shot setting; and iv) how useful are learnt time-consistent representations.

### 5.1 TASKS

We answer these questions based on four tasks with increasing complexity, implemented with MuJoCo (Todorov et al., 2012). We consider an attempt successful if the agent gets under a threshold distance

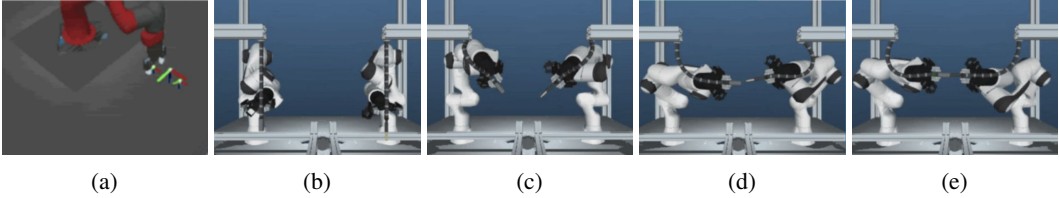

| (a) | (b) | (c) | (d) | (e) |

Figure 5: **Tasks description** a) is the parameterised reach task. Visiting the small yellow goals must happen before reaching the middle goal. b) is dual arm reaching, we combine this with c) lifting, d) aligning in both position and orientation and e) is the dual arm insertion.

from its target. The metrics include the running accuracy of completing the tasks and the overall accumulated reward. All policies are executed in 10Hz with 12 random seeds. At test time we sample a random goal from all successful goal states and for the agent. We use $z = (s, e(s))_t$ and $z_g = e(g)$, for all encoders $e(\cdot)$ and always train with a sparse binary environment reward. All methods use the same representation and rewards as HinDRL. The number $K$ of goal samples is dependent on the complexity of the task with further details in the next paragraphs and in Appendix A.10. We evaluate HinDRL on two different robotic set ups and a total of four different tasks. Each environment assumes a different robot which has different structure, action space and robot dynamics.

**Parameterised Reach:** Here, we use a single 7DoF Sawyer robot to perform parameterised reaching. Parameterised reaching is the task of visiting two randomly located in free space way-points before reaching for its target goal, Figure 5a. The continuous action space is comprised of the Cartesian 6 DoF pose of the robot and a 1 DoF representing the gripper open/close motion. The goal space is equivalent to the state space. We use a total of two hindsight goal samples, $z_g$, to relabel a single step.

**Bring Near:** This is a dual-arm task situated in a cell with two Panda Franka Emika arms and hanging audio cables, Figure 5b-c. The agent has to use both robot arms to reach for two cables and bring both tips within an $\varepsilon$ distance from each other. The action space is 7DoF for each arm, representing pose and grasp. The state and goal spaces are twice as large due to the dual nature of the task. Here we use a total of four hindsight goal samples, $z_g$, to relabel a single time step.

**Bring Near and Orient:** This is a dual-arm task similar to above where the agent needs to reach and grasp the two cables. However, here it is also required to align both tips, that is position and reorient them within a certain threshold. This task is significantly more complicated than the one above as it requires an additional planning component. The two cables have rigid components at their tips, a 3.5mm jack and a socket respectively. In order to be manipulated to a specific orientation those cables have to be grasped at the rigid parts as grasping the flexible part of the cable prevents from aligning the two tips (Figure 5d). We used a total of six goal samples, $z_g$, to relabel a single time step.

**Bimanual Insertion:** This is the most complex task we evaluate against. It has the same dual-arm set up as above but requires the complete insertion of the 3.5mm jack as well as the reaching, grasping and alignment stages from the previous two bimanual tasks (Figure 5e). This task requires not only careful planning but also very precise manipulation in order to successfully complete insertion. We used a total of 12 hindsight goal samples, $z_g$, to relabel a single time step for this task.

## 5.2 HINDSIGHT GOAL SELECTION FOR DEMO-DRIVEN RL: HINDRL

In this section we evaluate the utility of using different strategies for goal sampling in hindsight in the context of demo-driven RL against a hand-engineered encoder. We compare the performance of HinDRL against several strategies for hindsight goal sampling, as well as two non-goal-conditioned baselines – vanilla DPGfD and Behaviour Cloning (BC).

In addition, we combine DPGfD with hindsight relabelling using only the final reached state by the agent, similar to (Nair et al., 2018a) although without a curriculum over start configurations and using our goal-conditioned distributional agent as opposed to DDPG. We refer to this as HER (final). We consider using samples from the future trajectory to relabel both the demonstrations and the agent rollouts, similar to (Ding et al., 2019), although we do not consider using a stronger BC component and use our goal-conditioned distributional agent as opposed to DDPG. we refer to this baseline as HER (future). We report findings in Table 1. In all cases HinDRL performs best. Notably, assuming a sufficient number of demonstrations, the DPGfD agent is able to solve both the parameterised reach and the bring near tasks very well, but struggles on the full cable-insertion task. Both HER-based samplers struggle on this task as well, but HER (future) performs better. HER is particularly good

for shorter-horizon tasks like Bring Near. We do not expect to have any significant performance benefits against HER on such tasks. In summary, the ability of HinDRL to specialize at achieving goals on track to the specific task solution through the proposed task-constrained self-supervision results in a superior performance across all considered long-horizon tasks.

| Environment | BC Mean (Std) | DPGfD Mean (Std) | HER (final) Mean (Std) | HER (future) Mean (Std) | HinDRL (Our) Mean (Std) |
|---|---|---|---|---|---|
| Parameterised Reach (2wp) | 82.05% (0.0) | 88.62% (7.7) | 84.92% (28.6) | 86.92 (5.2) | **92.61% (4.7)** |
| Bring Near | 88.90% (0.0) | **99.74% (1.0)** | 32.03% (41.8) | 97.75% (2.9) | 98.79% (4.1) |
| Bring Near + Orient | 57.89% (0.0) | 83.52% (18.0) | 8.33% (27.6) | 23.28% (33.5) | **90.77% (10.0)** |
| Bimanual Insertion | 16.06% (0.0) | 4.28% (4.4) | 3.38% (3.3) | 18.39% (12.3) | **78.92% (10.3)** |
| Average | 61.23% (0.0) | 69.04% (7.8) | 32.17% (25.32) | 56.59% (12.9) | **90.27% (7.3)** |

Table 1: Records performance in terms of accuracy.

## 5.3 PERFORMANCE ON FEW-SHOT TASKS

| Demonstrations | BC Mean (Std) | DPGfD Mean (Std) | HER (future) Mean (Std) | HinDRL (Our) Mean (Std) |
|---|---|---|---|---|
| 55 demos | 16.06% (0.0) | 4.28% (4.4) | 18.39% | **78.92% (10.3)** |
| 10 demos | 2.94% (0.0) | 0.0% (0.0) | 0.0% (0.0) | **32.05% (23.3)** |
| 5 demos | 0.52% (0.0) | 0.0% (0.0) | 0.0% (0.0) | **17.78% (18.9)** |
| 1 demos | 0.13% (0.0) | 0.0% (0.0) | 0.0% (0.0) | **12.58% (18.3)** |

Table 2: Performance against using different number of demonstrations on Bimanual Insertion.

Using demonstrations can significantly speed up the training time while still successfully outperforming the expert eventually, (Vecerik et al., 2019). However, the final achieved performance in sparse reward settings depends on the complexity of the task and the number of provided demonstrations. In this section we show how using task-conditioned hindsight goal selection can significantly reduce the demonstration requirements and relax the dependency of the RL agent to the performance of the BC loss. We evaluate the ability of HinDRL to work in a few-shot setting. We train with 1, 5, 10 and 30 demonstrations and report our findings against the Bring Near + Orient task in Figure 6. The figure shows the overall achieved accuracy across different number of demonstrations. There, all solutions fail to solve the task with a single demonstration. This shows that all considered DPGfD-based solutions were dependent on the ability of the BC component to achieve more than 0% accuracy. HinDRL is able to outperform the BC policy rather quickly on the 5-and-10-shot tasks while both DPGfD and HER took significantly longer. Furthermore, HinDRL manages to maintain its final achieved accuracy across the 5, 10 and 30-shot cases for the dedicated training budget while both DPGfD and BC didn't. Table 2 shows the final achieved accuracy on the Bi-manual Insertion task. There, only HinDRL was able to learn in the 10-5-1-shot scenarios. This indicates the ability of task-constrained hindsight relabelling to bootstrap learning for long-horizon dexterous manipulation tasks, particularly with high dimensional continuous action spaces.

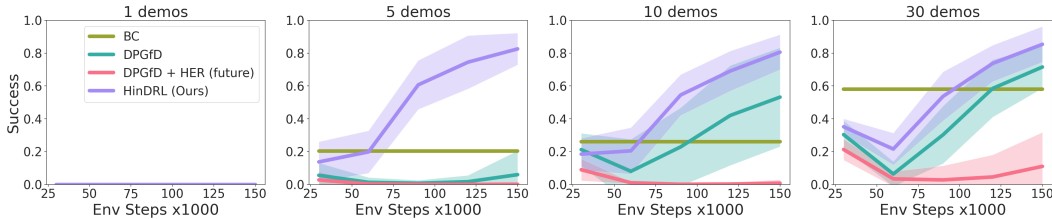

Figure 6: Accuracy against different number of demonstrations on the Bring Near + Orient task. Each plot shows the overall achieved accuracy during training. HinDRL consistently outperforms the alternatives. Success with one demonstration depends on the selected demonstration. We choose it on a random principle.

## 5.4 STUDYING THE SPEED OF EXECUTION

We study the dependency of HinDRL on the number of demonstrations and the achieved accuracy of the BC component. The results show a definite improvement over these two factors. However, this does not necessarily mean that HinDRL is strictly better than the expert's performance. In practice, a quick and nimble expert can still produce higher total output and be more valuable on average than an RL solution even if it is less successful. Therefore, in this section we study the overall speed of performing a task. We measure the speed of solving a task as a function of the per-step reward accumulated over the course of the entire training. Therefore, an increased per-step reward means

that an episode takes less environment steps to complete. We report our findings in Figure 7. We provide an additional ablation over using different numbers of demonstrations in Appendix A.6. Our findings indicate that HinDRL was always able to outperform the speed of execution obtained from the BC policy while using only DPGfD or DPGfD with HER did not. HER is beneficial for the easier tasks with insufficient number of demonstrations for DPGfD. However, HER alone is unable to solve the longer-horizon complex sequential tasks.

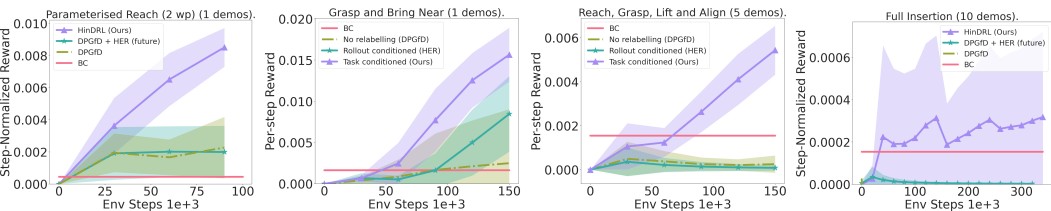

Figure 7: Measuring per-step reward using the smallest number of demonstrations that resulted in learning.

## 5.5 ROBUSTNESS TO THE QUALITY OF THE ENCODER

In the above sections we study performance when using different hindsight goal samplers, HinDRL sensitivity to the number of demonstrations and its ability to learn more efficient policies than the provided expert. However, in all those cases we assumed access to a hand-engineered state encoder. Informative representations is of central important to HinDRL as the algorithm uses the state representations both as input to the RL agent but also during the relabelling stage to assign reward, as discussed in Section 4. However, access to near perfect representations is not always feasible. Therefore, we study the sensitivity of our solution to the quality of the encoder. We evaluate the achieved performance against using learnt representations and raw input too. We compare learnt representations. One is extracted with a $\beta$-VAE, for $\beta = 0.5$, and one with TCC. We hypothesise that the notion of episode progress is of crucial importance to HinDRL. We report our findings in Figure 8. Representations that preserve temporal consistency were able to achieve higher results on all tasks. The dip in success rate is caused from annealing away the BC term in the actor's loss. Using a learnt encoding obtained with TCC was capable of achieving near equivalent performance to using an engineered encoding. We notice that using a support, $R_G$ for the goal distribution that is comprised of the union between the task-conditioned distribution and HER help improve the $\beta$-VAE performance, indicating that the agent's rollout was able to partially substitute the lack of notion of progress. However, we provide this more detailed ablation in Appendix A.5. Unlike the engineered encoding, the TCC-based representation does not contain any manipulation-specific priors.

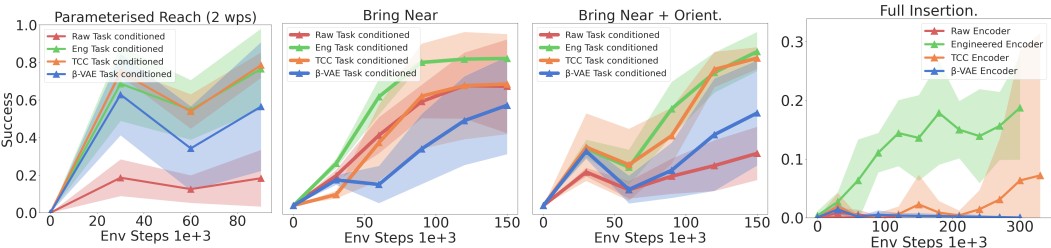

Figure 8: Robustness to the quality of the encoder. Comparing the engineered encoder with raw state observations and learnt $\beta$-VAE and TCC based encoders. The engineered encoder performs best with TCC achieving close to commensurate performance.

## 6 CONCLUSION

We proposed an efficient goal-conditioned, demonstration-driven solution to solving complex sequential tasks in sparse reward settings. We introduce a novel sampling heuristic for hindsight relabelling using goals directly from the demonstrations and combine it with both engineered and learnt encoders that consistently preserve the notion of progress. We show that HinDRL outperforms competitive baselines. Moreover, the method requires a considerably lower number of demonstrations. In the future, we would like to extend this work to multi-task settings, employ vision and move towards the context of batch RL, e.g. through employing distance learning techniques (Hartikainen et al., 2020) to build on more informative goal spaces.

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

# A APPENDIX

## A.1 PREDICTING PROGRESS

We measure the performance of the learnt representations to encode progress by running a KNN classification. First, we collect 500 demonstrations on the full insertion task using manually defined way points and a PD controller. We vary the starting configuration of each robot arm by $1°$ for all its 7 joints. Then, we perform a CV split over the collected trajectories and obtain training and validation sets. We train both encoders using the training data. Once training is complete, we process all training trajectories with each encoder which results in two separate encoded data sets. We use those to

| Model | Train | Test |
|-------|-------|------|
| VAE | **88.6%** | 63.6% |
| TCC | 86.2% | **79.2%** |

Figure 9: Predicting episode progress with 10-fold KNN.

train two separate 10-fold KNN classifiers - one for each type of encoding. Then, we process the never seen before validation set using each of the encoders and evaluate the accuracy of predicting the episode progress with the KNN classifiers. Table 9 shows the results. It can be seen that the VAE achieved much higher accuracy when evaluated on the training data as opposed to test, indicating it has overfitted to it. In contrast, the TCC was much better at predicting the episode progress.

## A.2 ABLATION OF THE ONLINE GOAL SELECTION STAGE

Goal-conditioned (gc) policy learning takes in as input a desired goal state to solve for. However, in cases where the goal is as abstract as 'solve the task' there may be a number of different goals describing the same problem. Therefore, restricting the gc policy to a single target goal state may negatively impact the learning process. Additionally, using a small subset of goals, e.g. goals corresponding to the final states of all demonstrations as done in Nair et al. (2018a), may be insufficient too. In contrast, conditioning on a wider range of goal states that solve the same task can better capture the goal distribution describing the task at hand. As a result, we randomly sample a plausible target goal state for each roll-out during training (see Figure 2). We propose to continuously grow a target goal distribution as training evolves and the agent starts solving the training task. In this section, we compare the performance of our

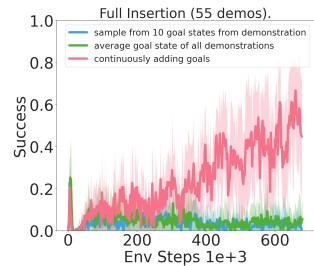

Figure 10: Different types of online goal selections.

agent when using a single target goal to represent solving the task, using as target goals only the goal states from the provided near-optimal demonstrations, and continuously growing the goal database as learning progresses. Our findings reported in Figure 10 show that continuously growing the goal database allows for better and faster learning in the context of vaguely formulated goals such as the 3.5mm jack insertion considered in this work.

## A.3 MIXING THE GOAL DISTRIBUTION SUPPORT

Mixing goal candidates taken from the agent's rollout and the provided demonstrations can help for retroactive goal selection. A potential scenario is the one discussed in Appendix A.4. Having noisy, sub-optimal representations that do not preserve the notion of progress can be problematic. An

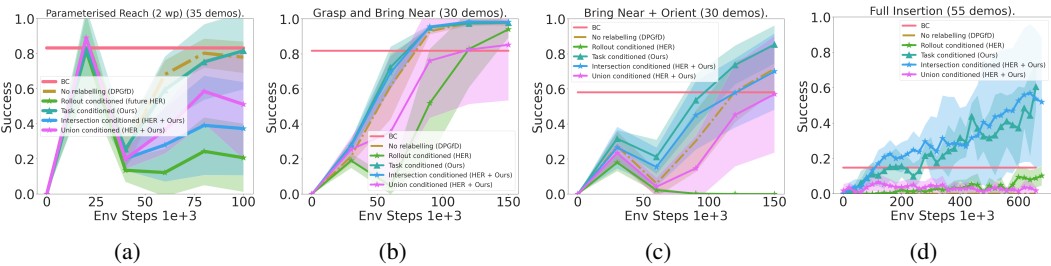

Figure 11: Measuring per-step reward using the smallest number of demonstrations that resulted in learning.

alternative scenario could involve a relatively narrow set of demonstrations. This is particularly evident in complex task settings where having a set of successful demonstrations can still be insufficient to solve the task as there are a vast number of failure modes, like in the full insertion task, for example. In this subsection, we focus on studying different candidate heuristics that can help relax these constraints. We build upon the formulation for retroactive goal selection introduced in Section 4.2 which allows us to fuse together the HER-style relabelling strategies and demo-driven ones too.

**Using demonstration states as candidate goals:** The simplest version of this is a union over both sets where the agent gets to sample a goal that comes from the rollout or the demonstration data. This can be expressed as $R_G = \{z_g \in \zeta \bigcup \mathscr{D} : p(z_g) > 0\}$. Such formulation can be useful, for example when the provided demonstrations are only partially useful to solving the task. Implicitly letting the agent to sample goals that are part of its own rollout can help model useful behaviours that over time could help get us closer to the provided demonstrations.

An alternative version is when the goal distribution $p(z_g)$ is composed of the intersection over the two types of goal distributions discussed in Section 4.2. In this case, the support of the goal distribution becomes $R_G = \{z_g \in \zeta \bigcap \mathscr{D} : p(z_g) > 0\}$. We find this intersection to be useful in cases where the adopted representation does not have an encoded notion of progress. Therefore, choosing goals that are $\varepsilon-$close to states produced by the agent's dynamics can hypothetically act as regularisation over the choice of goals we use but still ensure staying close to the target task. We can collect all qualifying goals for a time step $t$ by iterating over the data set of successful trajectories obtained from demonstration and compare to $z_t$. Since $z_t$ and all $z_g$ are temporally consistent, we can use Eq. 3 to prune the data set and pick the closest $z_g$ for each $z_t$. We used a task-conditioned sampler where we sample directly from a distribution implied from demonstrations. However, we can also compose distributions comprised of mixing goals from the agent's rollout and the demonstrations. Here we consider two different versions of this.

We report our overall results in Figure 11. Our results indicate that using the intersection over the rollout trajectory and the demonstrated goal results in broadly similar resutls as the task conditioned approach. However, the intersection based solution had much higher variance indicating that the agent is is less stable where some seeds achieved near perfect performance and others were closer to failure. We noticed that this type of goal conditioning can be useful when training with a VAE. That is, this sampling strategy can be useful in cases where the quality of the representations and also of the implied task distribution is poor, e.g. when they do not contain notion of progress or in low data regimes. We report these details in Appendix A.4.

**Using relevant agent states as candidate goals:**
Utilising the demonstration states to collect candidate goal distributions can be a powerful tool as we demonstrate in this work. However, in complex tasks, such as the full insertion task (Figure 1), relabeling the goals with just demonstration states can sometimes fail to make the sparse-reward problem easier (as intended by HER) since the resulting goal distributions are still relatively narrow. This is particularly true in the beginning of the training process when there is a relatively large mismatch between the agent's Q function and the true underlying dynamics the provided demonstrations follow.

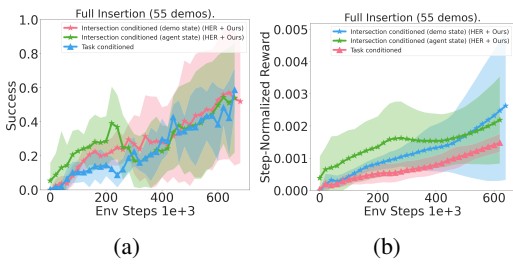

(a)         (b)

Figure 12: Comparing different demo-driven support distributions.

Therefore, we consider an alternative method that can help speed up the training process. To this end, we can form a collection of candidate goals that is jointly conditioned on both the agent and the demonstrator's behaviours but is comprised of all $z_t$ that fall under the $\varepsilon$ threshold defined in Appendix A.8. In this setting, we focus on modelling the agent's behaviour directly by focusing only on relevant to the task states as opposed to strictly targeting actual demonstration states. Figure 12 summarises our findings. While using this type of joint conditioning can speed up training (plot on the right), it does not necessarily result in improved performance (plot on the left). We suspect that mixing the goal distribution support can be potentially very useful to using less demonstrations or partially useful demonstrations. We leave this study for future work.

## A.4 NOISE SENSITIVITY OF THE ENCODER: CHALLENGING THE NOTION OF PROGRESS

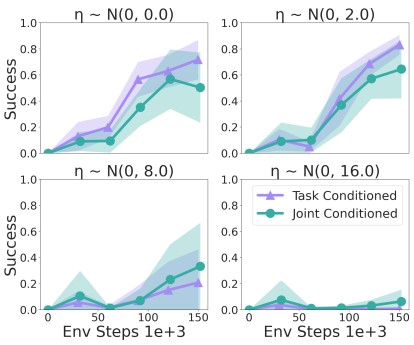

Figure 13: Injecting noise to the engineered encoder. Reach, Grasp, Lift, Orient Task.

We compare the performance of both the joint- and task-conditioned samplers on the Bring Near + Orient task and report the overall accuracy. We trained for a total of 150K environmental steps and report our findings in Figure 13. Relying on task-conditioned samples results in higher accuracy for the less noisy observations. This indicates that having a stronger representation is directly related to the agent's confidence in 'understanding' the actual task it has to solve. This study further confirms our conjecture that notion of progress is paramount to solving complex sequential tasks. Note that the relationship between the level of noise and performance depicted in Figure 13 does not affect the jointly conditioned relabelling strategies as much as it does for task-conditioned relabelling. In fact, a little bit of noise leads to improved performance for the former while it slows down training for the latter. This indicates that relying on alternative mechanisms for indicating progress, such as conditioning on the agent's own trajectory can be useful when progress is not successfully encoded in the representations used. This aligns with our motivation from Section 4 that task-conditioning works when we are confident in the quality of the task distribution. However, the proposed ablation in this section points towards an alternative mode that can potentially compensate for this. Namely, implicitly informing the agent for the notion of progress e.g. through building heuristics for hindsight goal selection that utilise both demonstrations and agent motion can be useful. We hypothesise that retroactive relabelling using only goals that are both similar to the demonstrations and aligned with the current agent's trajectory can be useful with respect to the agent's current understanding of the dynamics, represented through its current Q function. Next, we consider three different strategies for extracting candidate goals and discuss some of their benefits and limitations.

## A.5 QUALITY OF ENCODER: EXTENDED STUDY

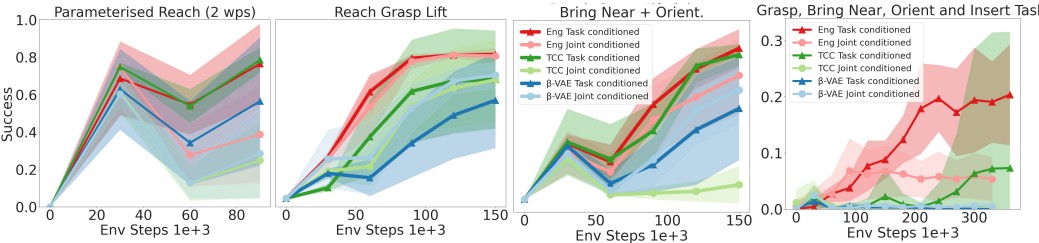

Figure 14: Learnt representations.

The previous section suggests that a joint-conditioned sampler can be more useful when the used representations do not encode notion of progress. In this section we compare using task-conditioned and joint-conditioned encoders using learnt representations instead. We compare using TCC and VAE and benchmark the results against a hand-engineered encoder. Even though TCC with a task-conditioned sampler achieved the closest results to the best hand-engineered solution across all tasks, we can see that a joint-conditioned encoder can work better for states that do not encode notion of progress.

Figure 14 illustrates the achieved results. We can see that using a $\beta$-VAE encoder worked best with goal sampling from a joint-conditioned goal distribution, $R_G = \{z_g \in \zeta \bigcap \mathscr{D} : p(z_g) > 0\}$. Note that the intersection between both distributions still results in a data set comprised of goal candidates that still belong to the implied from demonstrations task distribution. However, we only used the goals that were similar to the agent rollout. This result is connected to our observations from Appendix A.4 that a joint-conditioned sampler implicitly introduces a notion of progress via utilising the agent's own motion at the retroactive goal selection stage.

Another interesting observation is the final full insertion's task performance. Although TCC-based representation came closest to the engineered representation, it was still much lower. The full insertion task relies the most on the contact-rich manipulation to be completed when compared to the rest. The engineered representation contains information relevant to the manipulation which is why we suspect the gap between both learnt and engineered representation is much larger than the rest of the tasks. A potentially exciting future direction is attempting to extract representations that preserve the notion of contact as well as progress.

## A.6 PER-STEP REWARD

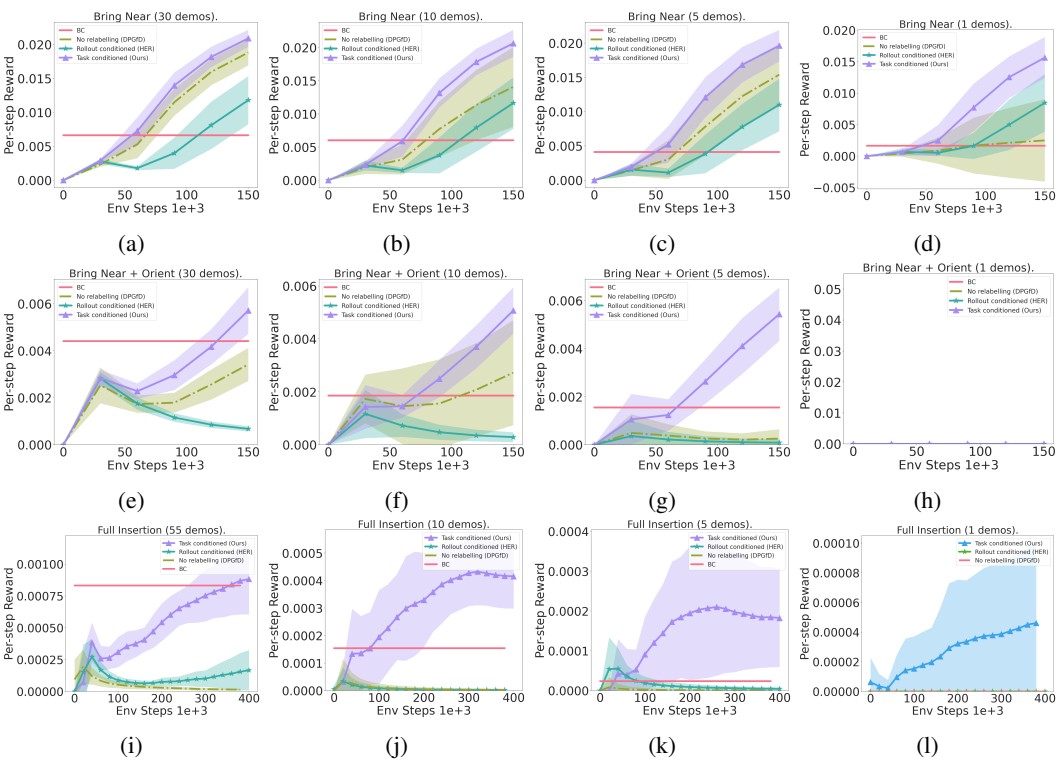

Figure 15: Measuring per-step reward.

## A.7 PROGRESS-BASED WEIGHTING

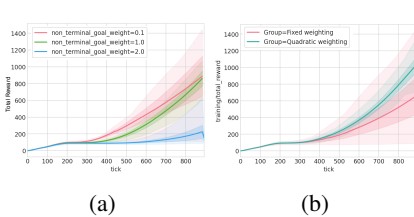

Figure 16: Measuring per-step reward using the smallest number of demonstrations that resulted in learning.

Weighting down the BC and actor-critic losses associated with intermediate states can have slight benefits to improving the speed of learning of goal-conditioned DPGfD. Although in practice there could be many different ways of reweighing loss values, we found two particular ones useful in our setting. One way of scaling such losses is by choosing a fixed weight $\omega$ that scales down all non-terminal states' losses during training by the same fixed weight. An alternative weighting can be defined by using a quadratically scaled weight using the episode progress. That is, for batch b, we get $\mathscr{L}_{BC}(b) = \lambda^p * \mathscr{L}_{BC}(b)$ and $\mathscr{L}_{TD} = \lambda^p * \mathscr{L}_{TD}$, where $*$ indicates element-wise multiplication and

$$\lambda^p = \begin{cases} 1.0, & \text{if } z_t = T \\ \omega, & \text{otherwise} \end{cases}, \text{ or } \lambda^p = i^2, \text{ for } i \in \{\frac{1}{T}, \dots, \frac{T}{T}\}. \tag{4}$$

We used $\omega = 0.1$ in our experiments. Figure 16 illustrates an example of re-weighting on the Bring Near + Orient task. Down-weighting intermediate states can lead to slightly faster learning and a

higher variance performance. There is a difference between weighting states using a fixed value and assigning quadratic weighting proportional to the episode progress.

## A.8 Computing the threshold

There are multiple ways to obtain an $\varepsilon$ threshold for our goal conditioned reward. We used the provided demonstrations to compute the average distance $\varepsilon = \mu + k\sigma$, where $\mu$ and $\sigma$ were extracted using a rolling distance between consecutive states from the encoded demonstrations, e.g. $||z_t^d - z_{t+m}^d||$, for a demonstration $d$ with an $m$ step gap in between the two states and $k$ standard deviations. In our tasks, $m = 10$ for all tasks but the bring near and orient and the full insertion where we used $m = 5$. We use a rolling distance over a window of time steps because we did not want to let time step clusters often situated around the different "narrow phases" of a trajectory influence the average threshold. Broadly, we notice a relationship between the size of the rolling window and the precision and recall of the obtained goal-conditioned sparse reward function. We ablate the importance of a rolling window size on the bring near

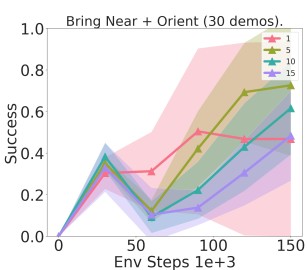

Figure 17: Ablating the rolling window.

and orient task in Figure 17. There, it can be seen that too small of a rolling window size (such as 1) might have a noticeable negative impact on learning due to the clusters situated around the different phases. Effectively, too small of a window can affect the recall of our obtained reward which can be detrimental to learning. In contrast, too large of a rolling window can affect the speed of learning due to allowing for a more flexible threshold function. Using too large of a rolling window can reduce the precision of the obtained threshold by rewarding too many false positive states. In terms of learning, this can be detrimental to the speed of learning a successful policy and might result in converging to poorer performance too.

## A.9 Computing the engineered encoders

The engineered goal encoders vary between tasks, but in all cases the encoding captures some notion of progress of the agent through the set task.

**Parameterized Reach:** The engineered encoder for this task concatenates the robot arm pose and a mask of which waypoints have been visited so far.

**Bring Near:** The state encoder is the concatenation of the distance of the left and right grippers from their corresponding cables, the distance between the cable tips, and of whether the grippers have grasped their respective cables.

**Bring Near and Orient:** The encoder for this task is similar to Bring Near, but also adds the dot product between the z-axes of the left and right cable tips.

**Bimanual Insertion:** The encoder for this task is similar to Bring Near and Orient, but adds the distance of the cable tip from the socket bottom, along the z-axis of the socket.

## A.10 Choosing number of hindsight goal samples

We followed the intuition that the complexity of the task guides the number of samples required to capture the overall task distribution. That is, we used 2 samples for the simplest task of Parameterised Reach, 4 for the Bring Near, 6 for Bring Near and Orient and 12 for the Bi-manual Insertion.

