# OpenReview forum: "Wish you were here: Hindsight Goal Selection for long-horizon dexterous manipulation"
_ICLR.cc/2022/Conference — ICLR 2022 Poster_

### Official Review · Reviewer_jLXD · 2021-11-01

**Correctness:** 4
**Technical Novelty And Significance:** 2
**Empirical Novelty And Significance:** 3
**Recommendation:** 6
**Confidence:** 3

**Main Review:**

Strong Points

+ The final task used in the paper (“bimanual insertion”) is challenging enough to convince readers of the authors’ claim that the proposed algorithm is a meaningful improvement on learning long-horizon control tasks. Authors’ breaking it down to several tasks by complexity is very helpful.

+ The empirical results are comprehensive and convincing. Difference between the proposed algorithm and prior works is big.

+ The paper is well motivated. The overall “storyline” is clear and makes sense.

Weak Points

- The amount of novelty is mediocre. The main innovation seems to be the new technique for hindsight relabeling.

- Technical clarity can be improved, especially in the way prior works are introduced in the Methodology section, which seems a bit high-level and requires readers to have substantial knowledge in the prior works to understand the full picture.

Additional Questions

? I’m curious to learn from the authors why HER (both the “final” and “future” variants) underperforms DPGfD in the Bring Near and Bring Near + Orient tasks, as shown in Table 1?


**Summary Of The Paper:**

This paper aims to improve the learning effectiveness and data efficiency for long-horizon control tasks under sparse rewards with demonstrations. Building upon prior works on DPGfD and HER, the authors proposed a new hindsight goal relabeling technique, which is referred to as “task-constrained goal-conditioned RL” to differentiate against “general goal-conditioned RL” as in prior works. Specifically, instead of selecting goals from the learned agent’s rollouts, the new technique only selects goals from the successful rollouts (and demonstrations). Rigorous experiments demonstrated that the proposed algorithm outperforms prior works on a very challenging bidextrous peg insertion task in simulation, in the aspects of the learned agent’s final performance (e.g. task completion rate) and data efficiency (e.g. number of demonstrations).

**Summary Of The Review:**

Overall I enjoy reading this paper. While the novelty is not steller, the experimental results are convincing and impressive. I recommend acceptance for broader dissemination.

---

> ### Author Response · Authors · 2021-11-13
> **Addressing clarity concerns and answering HER-related question**
>
> > ...the proposed algorithm is a meaningful improvement on learning long-horizon control tasks...results are comprehensive and convincing...the overall “storyline” is clear and makes sense.
>
> Thanks for recognising the value and clarity of our work. We have adapted the text and highlighted the relevant areas by colour coding and referencing replies using the reviewers unique ids. We hope that this will contribute to an easier and clearer discussion.
>
> > The amount of novelty is mediocre. The main innovation seems to be the new technique for hindsight relabeling.
>
> Thanks for probing the clarity of our work, we believe we have not made our technical novelty entirely clear.
>
> We propose a framework for efficiently solving long-horizon complex sequential tasks.
>
> Our framework introduces:
> - a novel technique for hindsight relabelling that allows for smoothly varying the task relevance of the relabelling process;
> - an improved way for describing abstract target tasks (as opposed to explicitly defined random goals) used in the online goal selection stage; and
> - An effective approach for utilizing time-consistent representations and distance-based rewards in goal-conditioned RL
>
> We update Section 1 to state this more clearly.
>
> > Technical clarity can be improved, especially in the way prior works are introduced in the Methodology section, which seems a bit high-level and requires readers to have substantial knowledge in the prior works to understand the full picture.
>
> Clarity of our work is of paramount importance to us. We would be very grateful if Reviewer jLXD shared which part of the introduction of prior works they found too abstract. We thank Reviewer jLXD in advance!
>
> > I’m curious to learn from the authors why HER (both the “final” and “future” variants) underperforms DPGfD in the Bring Near and Bring Near + Orient tasks, as shown in Table 1?
>
> Both DPGfD and HER target the exploration problem in sparse reward settings. The main difference is that DPGfD targets this explicitly through using auxiliary losses like BC by biasing the actor’s loss at the early stages of training, this leads to very fast skill acquisition. In contrast, HER addresses the exploration issue implicitly through self-supervision and thus learns more general policies than DPGfD but can take longer and it may also fail to scale to long-horizon tasks, as observed in concurrent works too, Gupta et al. 2019. Our results showed that naively combining the two approaches interfered with each other's performance regardless of the effort we put to mitigate this. While DPGfD can solve targeted tasks faster, combining it with HER reduces its capacity. In contrast, HER really picks up when BC was unable to help with the task, which is the case for the full insertion task. However, it did not solve the task. We tried removing the BC loss from the agent too, effectively training DPG + HER. However, given the limited budget and the complexity of the tasks, this agent performed even worse than DPGfD + HER.

---

> > ### Comment · Reviewer_jLXD · 2021-11-28
> > **Response to authors**
> >
> > I thank the authors for their response, which helps me better understand the work. But I don't think a change in the recommendation score I originally gave is warranted at this point.

---

### Official Review · Reviewer_rSZQ · 2021-11-02

**Correctness:** 2
**Technical Novelty And Significance:** 3
**Empirical Novelty And Significance:** 2
**Recommendation:** 6
**Confidence:** 3

**Main Review:**

The strong points of this paper are:

A more comprehensive ablation study of topics such as encoder quality, effect of number of demonstrations on method performance, and goal-distribution sampling, which are often important but unclear aspects of implementing these types of goal-conditioned methods.

The use of TCC to learn the latent observation representations seems quite promising as shown in Figure 4 and useful for the reward computation

The bimanual manipulation task which is evaluated on appears quite challenging and the method achieves significant gains on the shown tasks.

The weak points of this paper are:

I’m not sure if the claim made in the Related Literature section “However, in all those works, hindsight goals were always chosen directly from the agent’s own trajectory” is accurate -- Nair (2018a) also selects goals directly from demonstration states. Using demonstration states as relabeled rewards in itself does not seem novel.

The engineered encoder representation which is used for the method evaluation seems to essentially provide a more dense reward for the agent, which is the $\ell_2$ distance between the engineered features. In this case, I would argue that comparisons should be made to HER and other relabeling strategies which are given access to this engineered reward function (this may already be the case, but it was not clear to me when reading the text). When moving to use the TCC-based encoder and reward function, the performance on some tasks drops significantly, which leads me to believe that the performance improvements could largely be due to essentially giving more signal from the human expert.

Additional baselines would make the empirical results more convincing, for example, to Reinforcement Learning with Imagined Goals (RIG).

Additional questions:

How are the number of relabeled goals for each task selected?

Why is it that in Figure 9, the performance of the green line for full insertion is around 0.2, whereas it is much higher in Table 1? Are these the same method, or is the difference in number of training steps?

A few recommendations for the manuscript:

The reference to Table 6 in the text (results) should be to Table 1 instead? There seems to be no Table 6

Section 4.2 was rather hard to parse, for example I don’t quite understand how “In contrast, hindsight goal selection is the process of sampling candidate goal ....” Is different from “Instead, we sample goals for each time step from trajectory ζ using some candidate goal distribution p(zg)”.



**Summary Of The Paper:**

This paper contributes a method called HinDRL, which tackles the important problem of sparse reward robotic RL tasks when supplied with demonstrations. HinDRL seeks to build off of hindsight relabeling to learn a goal-conditioned policy which is trained in a self-supervised manner. Specifically, by relabeling transitions with task-specific goal candidates from expert demonstrations, the agent will be given more task-relevant feedback. By combining the goal selection strategy with goal-conditioned rewards which are either engineered or learned through temporal time-consistency, HinDRL is demonstrated on simulated sparse-reward robotics tasks where it achieves improved final performance and demo efficiency compared to HER.


**Summary Of The Review:**

My recommendation currently is to reject the paper, because I feel that the experimental results seem to indicate that the engineered features are contributing to the difference in performance and thus creating an unfair comparison, and the novelty of the goal selection method is not currently apparent to me. If the authors would be able to clarify either or both of these points, it would help immensely. If the methods which are compared to are not being currently given the engineered encodings, comparisons to versions which are given that information (as well as additional baselines) would be helpful.

---

> ### Author Response · Authors · 2021-11-13
> **Addressing fairness and novelty concerns**
>
> > A comprehensive ablation study ... the use of TCC ... seems quite promising ... and useful ... the bimanual manipulation task ... appears quite challenging and the method achieves significant gains on the shown tasks.
>
> Thank you for recognising the thoroughness, potential and capabilities of our work. We are pleased to see you find it promising! We have adapted the text and highlighted the relevant areas by colour coding and referencing replies using the reviewers unique ids. We hope that this will contribute to an easier and clearer discussion.
>
> > claim made in the Related Literature section is not accurate -- Nair (2018a) also selects goals directly from demonstration states.
>
> **summary**: Please note that Nair et al. (2018a) does not select goals directly from demonstration states during the hindsight goal selection stage. We realise we have not made this distinction clear enough so we update Section 2 to be clearer on this, add more details in Appendix A.2 and provide more details next.
>
> **more details**: There are two separate goal selection stages in both ours and Nair et al. (2018a) works. Namely, the online goal selection stage (stage A) and the hindsight goal selection stage (stage B). In our work, we differ from Nair et al. (2018a) in the way we do both stages. However, the cited sentence above refers to stage B so we will focus our reply on stage B only so that we avoid any potential confusion.
>
> During stage B, Nair et al. (2018a) select goals directly from the agent's rollout. Specifically, for every episode that their agent experiences, they store it in the replay buffer twice: once with the original goal pursued in the episode (as typically done in standard off-policy RL) and once after relabelling with the goal corresponding to the final state achieved in the agent's episode. This is stated explicitly in Section III.D of Nair et al. (2018a). Please note that in their hindsight relabelling stage, they do not use demonstrations.
>
> This is different from what we do. In HinDRL, for every episode that our agent experiences, we store it in the replay buffer K+1 many times (and not just twice): once with the original goal pursued in the episode (as typically done in standard off-policy RL) and K more times using goals uniformly sampled from all demonstration states. The key difference here is that during the hindsight relabelling stage, Nair et al. (2018a) selects the goal corresponding to the final state achieved in the agent's episode (we refer to this as HER (final) in our evaluation) while we sample goals from a task distribution obtained from demonstrations. Note that in our set up the task distribution contains goals along the entire trajectories of all demonstrations and not just their final states too.
>
> We would also like to draw Reviewer rSZQ's attention to Table 1 of our paper. There, we compare against the same hindsight sampling strategy used in Nair et al. (2018a), we refer to it as ''HER (final)''. It can be seen that this hindsight goal selection stage does not achieve the same performance as ours even though we train HER (final) using the exact same encoding and rewards as HinDRL.
>
> In addition, we included additional details on how we differ from Nair et al. (2018a) in terms of stage A and provide more experimental comparisons in Appendix A.2.
>
> > The engineered encoder representation ... seems to essentially provide a more dense reward for the agent ... comparisons should be made to HER and other relabeling strategies which are given access to this engineered reward function (this may already be the case, but it was not clear to me when reading the text).
>
> This is already the case. We use the same encoder and rewards for all baselines. We make this clearer in the updated paper, Section 5.1.
>
> > … the TCC-based encoder and reward function, the performance on some tasks drops significantly … the performance improvements could largely be due to the human expert.
>
> **sumary**: This is a great observation and it is correct. Still, the self-supervised representation obtained by TCC did significantly better than $\beta$-VAE which showcases the importance of episode progress to solving complex sequential tasks. We would also like to point out that Appendix A.5 of the updated paper discusses a version of HinDRL that works well using $\beta$-VAE representations too. We make referencing to A.5 clearer in the paper, Section 4.3.
>
> **more details**: We recognise that TCC representations might not always perform on-par with hand-engineered encoders but are certainly a big step forward towards understanding and solving long-horizon complex sequential tasks. Unlike the hand-engineered representation, TCC does not encode manipulation-specific information and we believe that this drop in performance was due to this. We see the problem of fusing temporally-consistent representations with manipulation-specific information with self-supervision as an interesting opportunity for future work.

---

> > ### Author Response · Authors · 2021-11-13
> > **Addressing RIG as a baseline and additional questions**
> >
> > > Additional baselines would make the empirical results more convincing, for example, to Reinforcement Learning with Imagined Goals (RIG).
> >
> > Thank you for consistently probing the clarity of our work and helping us make a more convincing and clear paper. RIG does not use demonstrations to bootstrap policy learning in their set up and previous literature shows that bootstrapping RL from demonstrations (RLfD) generally puts alternative approaches at a huge disadvantage (Vecerik et al. 2019). Directly comparing against RIG neglects their contributions which are orthogonal to our work. Instead, we compare only against the relevant aspects of the RIG architecture by inducing them in an RLfD setting. That is, we compare against using a $\beta$-VAE as an encoder and against the hindsight relabelling strategy RIG employs too.
> >
> > We realise we may not have made this connection clear enough before. We fix this with a more explicit referencing in Section 4.3. We would also like to draw Reviewer rSZQ's attention to Table 1. There, HER (future) refers to the same hindsight relabelling strategy as the one employed in RIG. Similarly, Section 5.5 and Appendix A.5 compare using representations obtained with a $\beta$-VAE with TCC. We hope that Reviewer rSZQ can agree with us that we have thoroughly evaluated all relevant components from RIG in the context of RL from demonstration.
> >
> > > How are the number of relabeled goals for each task selected?
> >
> > Those were arbitrarily selected following the intuition that harder tasks may require more samples per episode to capture the underlying task distributions. Therefore, we used 2 for the parameterised reach, 4 for the bring near, 6 for the bring near and orient and 12 samples for the full insertion task. We have added an additional subsection in Appendix that gives these details.
> >
> > > Why is it that in Figure 9, the performance of the green line for full insertion is around 0.2, whereas it is much higher in Table 1? Are these the same method, or is the difference in number of training steps?
> >
> > These are the same method but the difference is in the number of training steps.
> >
> > > The reference to Table 6 in the text (results) should be to Table 1 instead? There seems to be no Table 6
> >
> > We fixed this in the updated paper.
> >
> > > Section 4.2 was rather hard to parse, for example I don’t quite understand how “In contrast, hindsight goal selection is the process of sampling candidate goal ....” Is different from “Instead, we sample goals for each time step from trajectory ζ using some candidate goal distribution p(zg)”.
> >
> > **sumary**: We rewrote the section. We hope the new version is easier to parse. We will clarify the raised question next.
> >
> > **more details**: Unlike the online goal selection stage (stage A), the hindsight goal selection stage (stage B) is the process of retroactively sampling candidate goals after an episode rollout.
> >
> > _In contrast to stage A, the sampled goals from stage B are not conceptually related to the abstract formulation of a target task._ This is a subtle difference, candidate goals from stage B do not represent abstract concepts like insertion but instead describe different stages from the behaviour that results in solving the abstract task. This is a subtle but important difference.
> >
> > _The way the hindsight goal selection stage works is that we sample goals for each time step of the trajectory_ $\zeta$ _using some candidate goal distribution_ $p(z_g)$ _implied from some given behaviour. Typically, this results in sampling candidate goals from the future states in rollout trajectories._
> >
> > > Addressing Reviewer rSZQ’s summary.
> >
> > We thank Reviewer rSZQ for the detailed analysis and constructive feedback provided for our work. We hope to have successfully answered the reviewer's concerns. Specifically, we hope that:
> >
> > - we successfully make it explicitly clear that we use the same encoders and rewards across all methods and thus we ensure a fair and consistent comparison against all baselines.
> > - it is now clear how HinDRL differs from the referenced work by Nair et al. (2018a) in terms of hindsight goal selection
> > - we also hope to have successfully convinced Reviewer rSZQ of the thoroughness of baseline coverage we provide in our work too.
> >
> > Finally, we would like to encourage reviewer rSZQ to ask us any further questions they may have. Their feedback has enormously helped us so far. Thank you.

---

> > > ### Comment · Reviewer_rSZQ · 2021-11-18
> > > **Response to authors**
> > >
> > > I would like to thank the authors for their thorough responses to my questions, and the clarifications which they have provided which have helped immensely. The clarification was quite helpful for the points about the encoder usage for baseline comparison, the differences between Nair et al. (2018a), and the baseline coverage compared to relevant components of RIG. This makes me more convinced of the effectiveness of the relabeling strategy based on the empirical evaluation. I have revised my score accordingly.

---

### Official Review · Reviewer_BCwv · 2021-11-02

**Correctness:** 4
**Technical Novelty And Significance:** 3
**Empirical Novelty And Significance:** 2
**Recommendation:** 6
**Confidence:** 4

**Main Review:**

Strengths:
- The paper studies a difficult problem in robotic manipulation, and shows results on tasks of varying difficulty. Leveraging demonstrations as task-relevant goals is intuitive and straightforward.

- The experiments are thorough. They study the importance of the goal selection strategy, sensitivity to different numbers of demonstrations, and how quickly different learned policies can solve the task.

Weaknesses:
- In the hindsight goal selection, relabeling the rewards for goals not part of the rollout trajectory necessitates a different reward function separate from the environment reward. The separate reward function seems to need to follow a sparse-reward structure to learn a Q-value function from both reward functions. This requires choosing a thresholding value \epsilon (although Appendix A.8 presents a seemingly general way to define what this should be, it ultimately still depends on task-specific knowledge of how large the window should be, from which \epsilon is computed). It would be nice to include a sensitivity analysis of this hyperparameter.

- Relabeling the goals with just demonstration states can fail to make the sparse-reward problem easier (as intended by HER) if the resulting goal distribution is still relatively narrow. For example, we could imagine tasks with a wide initial state distribution and a large state space that the demonstrations only sparsely cover, and even reaching demo states can be difficult. It seems like mixing the demo-driven goal distribution and distribution over reached states could lead to some benefits as shown in Appendix A.4.

- The choice of encoder seems to be quite important, with the TCC results being the closest to those with the hand-engineered features across the four tasks.

Finally, some of the implementation/experimental details are currently a bit unclear to me. My questions are below:

- Which encoder is used for the results in Table 1? Do the other methods in Table 1 operate over the same latent representation as HinDRL? Is the strong performance due to the proposed relabeling scheme or its combination with the extracted representation?

- What is p(z_g) in the definition of R_G? Is this derived from the demonstration data, e.g., a uniform distribution over the set of demonstration states?

- What does the “number of hindsight goal samples” hyper-parameter refer to? Is it the total number of hindsight goals, i.e., |R_G|?

- In Figure 7, why is there a dip in success rate for the different methods at around 60K environment steps? Is this because the BC term in the actor’s loss is annealed away?

Minor comments:
- The placement of the figures is a bit awkward. For example, Figure 8 corresponds to Sec. 5.4 but is placed in Sec. 5.5; Table 6 corresponds to Sec. 5.2 but is placed in Sec. 5.3. Some of the figures also appear in the paper out of order.

- It’d be nice to have a consistent legend across the 4 plots in Figure 12.

**Summary Of The Paper:**

This paper studies long-horizon manipulation tasks given sparse rewards and a few demonstrations from a hindsight relabeling-based approach. Specifically, it leverages the demonstration states as relevant goals in the goal relabeling process to guide task-specific exploration.

When evaluating the policy, the online goals are sampled from a goal dataset, composed of successful states and final states from the demonstrations. When relabeling the goals for training, the goals are sampled from the set of states visited by the demonstrations. Rather than operating over the raw state observations, this method utilizes an encoder, which can be either engineered by an expert or learned through self-supervision, to extract a latent representation. Then, the rewards are relabeled in hindsight based on a thresholded distance in the latent space.

**Summary Of The Review:**

The paper tackles an important and relevant problem: solving long-horizon tasks given a few demonstrations. The experiments are quite thorough, but there are some missing details that seem critical to evaluating the performance of the method compared to prior work. Also, it would be good to include a discussion of scenarios in which this goal relabeling method might fail, i.e., when reaching the demonstration states is difficult and so relabeling still doesn’t provide additional learning signal.

---

> ### Author Response · Authors · 2021-11-13
> **Addressing weakness points and more discussion over drawbacks**
>
> > … studies a difficult problem ... the approach is intuitive and straightforward. The experiments are thorough ...
>
> Thanks for recognising the clarity and thoroughness of our work and for the valuable feedback. We have adapted the prose and highlighted the relevant areas by colour coding and referencing replies using the reviewers unique ids. We hope that this will contribute to an easier and clearer discussion.
>
> > It would be nice to include a sensitivity analysis over the size of the window used for defining a thresholding value $\epsilon$.
>
> We are working on it and will add this soon!
>
> > Relabeling goals with demonstrations can sometimes fail ... mixing the two distributions could lead to some benefits as shown in Appendix A.4 ..  it would be good to include a discussion of scenarios in which this goal relabelling method might fail
>
> **summary**: We realise we have not clearly discussed the limitations of using demo-driven samplers and HER-style samplers even though we conduct rigorous experimentation studying their benefits. We add more details in the main part of the paper in Section 4.2 discussing edge cases and potential limitations. In addition, we extend our discussion in appendix and experiment with a few different ways to mitigate these limitations. We provide an in-depth analysis in Appendix A.3 of the updated paper (or A.4 in the review above). In summary, using demo-driven samplers can speed up training but it requires certain level of confidence over the quality of the task representations. This can sometimes be problematic as rightfully pointed out by Reviewer BCwv.
>
> **more details**: Using the agent's trajectory $\zeta$ for hindsight relabelling (as introduced in HER and follow-up works) is useful when modelling the agent's behaviour. However, it can struggle to scale beyond normal-horizon tasks, an observation that is also discussed in Gupta et al. 2019. On the other hand, using demonstration-driven relabelling heuristics can speed up training by directly modelling the task distribution using the provided demonstrations. However, this requires a certain level of confidence over the quality of the task representation. These two types of support distributions (defined with $\zeta$ and direct demonstrations) can complement each other and do not need to be disjoint. Mixing them up can be particularly useful in long-horizon cases where using just demonstration states fails to make the sparse reward problem easier, e.g. when we have insufficient demonstrations or the representations fail to capture the notion of progress. We provide an ablation against mixing samplers in Appendix A.3, including using the union, $\zeta \bigcup \mathcal{D}$, and intersection $\zeta \bigcap \mathcal{D}$ of the two sets of goals. Our results indicate that doing so can result in improving learning for noisy representations and even faster learning for harder tasks, such as the full insertion task.
>
> > The choice of encoder seems to be quite important, with the TCC results being the closest to those with the hand-engineered features across the four tasks.
>
> This is a great observation and it is correct. The self-supervised representation obtained by TCC did significantly better than $\beta$-VAE which showcases the importance of episode progress to solving complex sequential tasks. We would also like to point out that Appendix A.5 of the updated paper discusses a version of HinDRL that works well using $\beta$-VAE representations too. We make referencing to A.5 clearer in the paper, Section 4.3. The key observation we would like to make with respect to this result is that the notion of progress (e.g. through temporal consistency) is a very useful property for long-horizon complex sequential tasks.
>
> We see this as a valuable contribution to knowledge that helped us design our framework and can potentially be useful to future work too. Learning valuable inductive biases can help speed up learning in complex sequential robotics tasks and we argue that the notion of progress is a key ingredient.
>
> We add discussions in Appendicies A.4-5 and reference them in the main body of the updated paper in Section 4.2.

---

> > ### Author Response · Authors · 2021-11-13
> > **Addressing missing details**
> >
> > > Which encoder is used for the results in Table 1?
> >
> > We used the hand-engineered encoder in Table 1. We make this clearer in Section 5.2.
> >
> > > Do the other methods in Table 1 operate over the same latent representation as HinDRL?
> >
> > All methods always operate over the same latent representation and rewards as HinDRL. We make this clearer in Section 5.1.
> >
> > > Is the strong performance due to the proposed relabelling scheme or its combination with the extracted representation?
> >
> > We believe that it is due to the proposed relabelling scheme since all other methods have access to the same representations and rewards. We update the paper with these clarifications too, Section 5.1.
> >
> > > What is p(z_g) in the definition of R_G? Is this derived from the demonstration data, e.g., a uniform distribution over the set of demonstration states?
> >
> > This is correct, it is a uniform distribution over the set of demonstration states. We updated Section 4.2 to clarify this.
> >
> > > What does the “number of hindsight goal samples” hyper-parameter refer to? Is it the total number of hindsight goals, i.e., |R_G|?
> >
> > Yes, that is correct. We add an additional section in appendix, A.10 that describes this more clearly and provides guidelines for the choice of a total number. We also reference this explicitly in Section 5.1 of the updated paper.
> >
> > > In Figure 7, why is there a dip in success rate … around 60K environment steps? Is this because the BC term in the actor’s loss is annealed away?
> >
> > This is also correct, it is because the BC term in the actor's loss is annealed away. The prevalent RL loss results in a sudden drop in performance due to its exploratory nature. We clarify this in Section 5.5.
> >
> > > The placement of the figures is a bit awkward … Some of the figures also appear in the paper out of order. It’d be nice to have a consistent legend across the 4 plots in Figure 12.
> >
> > We have addressed these in the updated paper.
> >
> > > there are some missing details that seem critical to evaluating the performance of the method compared to prior work.  Also, it would be good to include a discussion of scenarios in which this goal relabelling method might fail...
> >
> > We thank Reviewer BCwv for the detailed analysis and constructive feedback provided for our work. We hope to have successfully answered the reviewer's concerns. Specifically, we hope that:
> >
> > - we successfully address all the pointed out missing details;
> > - make it explicitly clear that we ensure a fair and consistent comparison with all baselines;
> > - we provide a thorough discussion of scenarios in which this goal relabelling method might fail and convincingly propose ways to mitigate some of these limitations with our extended analysis.
> >
> > Finally, we would like to encourage reviewer BCwv to ask us any further questions they may have. Their feedback has enormously helped us so far. Thank you.

---

> > > ### Author Response · Authors · 2021-11-17
> > > **Addressing the sensitivity analysis over the size of the window used for defining a threshold**
> > >
> > > > It would be nice to include a sensitivity analysis over the size of the window used for defining a thresholding value $\epsilon$.
> > >
> > > We provide a sensitivity analysis and discuss it in Appendix A.8 of the updated paper. Here's a summary.
> > >
> > > Broadly, we notice a relationship between the size of the rolling window and the precision and recall of the obtained goal-conditioned sparse reward function. Our ablation shows that too small of a rolling window size (such as 1) might have a noticeable negative impact on learning due to the clusters of time-steps situated around the different phases. Effectively, too small of a window can affect the recall of our obtained reward which can be detrimental to learning. In contrast, too large of a rolling window can affect the speed of learning due to allowing for a more flexible threshold function. Using too large of a rolling window can reduce the precision of the obtained threshold by rewarding too many false positive states. In terms of learning, this can be detrimental to the speed of learning a successful policy and might result in converging to poorer performance too.
> > >
> > > We hope this addresses all Reviewer BCwv's concerns. We would like to also encourage the reviewer to let us know if they spot any additional weaknesses or have further questions. We appreciate their constructive feedback.

---

> > > > ### Comment · Reviewer_BCwv · 2021-11-20
> > > > **Thanks for the response**
> > > >
> > > > Hi, thank you for the detailed response and new sensitivity analysis. It’s addressed my concerns, especially about whether the comparisons operate over the same representation, so I’ve raised my score.

---

### Official Review · Reviewer_qVKc · 2021-11-03

**Correctness:** 3
**Technical Novelty And Significance:** 3
**Empirical Novelty And Significance:** 2
**Recommendation:** 6
**Confidence:** 4

**Main Review:**

Strengths:

This paper presents a (to my knowledge), novel goal-conditioned RL method. This approach improves appon Hindsight Experience Replay by incoporating demonstrations and efficiently uses them to set goals. I think this way of using goals will be quite useful for goal-conditioned RL in general. The method is well presented and easy to understand. The results are also well analyzed and HindRL shows strong performance when compared to HER and other RL + LfD approaches. I think the ablations, especially looking at the number of demonstrations needed for each method, are very detailed and provide good intuition about HindRL.


Weaknesses:

I think the main weakness of this paper is the lack of diversity in the empircal evaluations. It would be good to see results on more complex continuous control tasks in different settings than the bimanual cable insertion, for example the tasks from the HER paper (Andrychowicz et al., 2017). I am willing to increase my recommendation score if the authors can address this issue.



**Summary Of The Paper:**

The authors present a goal-conditioned RL method, which performs relabelling given a set of demonstrations. The main claim is that demonstrations will help guide exploration a lot more efficiently. This method is inspired from HER + LfD methods. The agent starts with a database of demonstrations. At training time, the agent samples a goal from this database, which it updates with successfully reached goal-states. In the hindsight relabelling process, the goals are sampled from the trajectory or from the demonstration database. This approach is evaluated on a set bimanual cable insertion tasks, where it outperforms prior approaches such as HER and is able to learn with fewer demonstrations than RL + LfD.

**Summary Of The Review:**

The method is novel, useful and well presented, but the paper lacks diversity in the domains evaluated.

---

> ### Author Response · Authors · 2021-11-13
> **Addressing diversity concerns**
>
> > ... a novel goal-conditioned RL method ... this way of using goals will be quite useful for goal-conditioned RL in general ... well presented and easy to understand.
>
> Thank you for recognising the novelty and utility of our work and for the valuable feedback! We have adapted the text and highlighted the relevant areas by colour coding and referencing replies using the reviewers unique ids. We hope that this will contribute to an easier and clearer discussion.
>
> > I think the main weakness of this paper is the lack of diversity in the empircal evaluations... add tasks from (Andrychowicz et al., 2017)
>
> Our paper is particularly attacking long horizon tasks while the environments in (Andrychowicz et al., 2017) are short horizon tasks. In short horizon tasks (potentially with less sparse rewards) we expect our method to perform similarly to HER and to not introduce any additional benefits.
>
> This can already be seen in simpler tasks presented in our paper. Specifically, the grasp and bring near task we use is an example of a shorter horizon task. There, the achieved results are comparable between HER and HinDRL. However, HinDRL outperforms HER significantly on all long-horizon tasks. We updated our paper to clearly state our targeted problematic and highlight that we do not expect additional benefits to HER in short-horizon tasks and make this clearer in Section 5.2. We also make it clearer that we evaluate on two different robotic set ups - single Sawyer arm and dual Panda Franka arm. Each environment assumes a different robot which has different structure, action space and robot dynamics. This is further clarified in Section 5.1 of the updated paper.
>
> We hope the provided clarifications convince Reviewer qVKc that we do not aim to claim superior performance against HER on shorter-horizon tasks and instead we target our contribution at solving long-horizon complex continuous control tasks only. Nevertheless, we welcome any additional feedback on why Reviewer qVKc sees the 4 tasks and 2 different robot setups we study our approach on as not diverse enough.

---

> > ### Comment · Reviewer_qVKc · 2021-11-29
> > **Diversity in tasks**
> >
> > Dear Authors,
> >
> > Thank you for your response! While I agree that four tasks on the dual arm setups are diverse from a task perspective they are still on a very similar setup. Reaching, aligning and insertion are all part of the same general task of cable manipulation. The second robotic does not have the same complexity, thus I am not sure if it is long horizon. While these clarifications are useful, I am not convinced that the method was evaluated in a diverse enough way.
> >
> > Best regards

---

### Decision · Program_Chairs · 2022-01-20

**Decision:**

Accept (Poster)

**Comment:**

This paper proposes a method to improve the sample efficiency of the HER algorithm by sampling goals from a distribution that is learned from human demonstrations. Empirical results on a simulated robotic insertion task show that the proposed method enjoys a better sample efficiency compared to HER.

The reviewers find the paper well-written overall and the proposed idea reasonable. However, there are concerns regarding the limited novelty of the proposed method, which seems incremental. Also, the empirical evaluation suffers from a lack of diversity. The considered tasks are virtually all equivalent to an insertion task. The paper would benefit from further empirical evaluations that include tasks such as those considered in the original HER paper.